# Effects of Corilagin on Lipopolysaccharide-Induced Acute Lung Injury via Regulation of NADPH Oxidase 2 and ERK/NF-*κ*B Signaling Pathways in a Mouse Model

**DOI:** 10.3390/biology11071058

**Published:** 2022-07-14

**Authors:** Fu-Chao Liu, Chia-Chih Liao, Hung-Chen Lee, An-Hsun Chou, Huang-Ping Yu

**Affiliations:** 1Department of Anesthesiology, Chang Gung Memorial Hospital, Linkou Branch, Taoyuan 333, Taiwan; ana5189@cgmh.org.tw (F.-C.L.); m7147@cgmh.org.tw (C.-C.L.); m7079@cgmh.org.tw (H.-C.L.); f5455@cgmh.org.tw (A.-H.C.); 2College of Medicine, Chang Gung University, Taoyuan 333, Taiwan

**Keywords:** corilagin, lipopolysaccharide, acute lung injury, inflammation, oxidative stress, NOX2, MAPK, NF-*κ*B

## Abstract

**Simple Summary:**

Corilagin has anti-inflammatory and antioxidant properties. Acute lung injury is a life-threatening disease. This investigated the protective effects and mechanisms of corilagin on acute lung injury. The results show that corilagin can reduce acute lung injury via attenuation of pro-inflammatory mediators and oxidative stress. Corilagin has potential utility as a therapeutic agent for the treatment of this life-threatening respiratory disease.

**Abstract:**

Acute lung injury (ALI) and acute respiratory distress syndrome are clinically life-threatening diseases. Corilagin, a major polyphenolic compound obtained from the herb *Phyllanthus urinaria*, has anti-inflammatory and antioxidant properties, and in this study, we sought to evaluate the protective effects and mechanisms of corilagin on lipopolysaccharide (LPS)-induced ALI in mice. ALI was induced in the mice by the intratracheal administration of LPS, and following 30 min of LPS challenge, corilagin (5 and 10 mg/kg body weight) was administered intraperitoneally. At 6 h post-LPS administration, lung tissues were collected for analysis. Corilagin treatment significantly attenuated inflammatory cell infiltration, the production of pro-inflammatory cytokines TNF-α, IL-6, and IL-1β, and oxidative stress in lung tissues. In addition, corilagin inhibited the LPS-induced expression of NOX2, ERK, and NF-*κ*B. Corilagin has anti-oxidative and anti-inflammatory effects, and can effectively reduce ALI via attenuation of the NOX2 and ERK/NF-*κ*B signaling pathways.

## 1. Introduction

Acute lung injury (ALI) and its most severe form, acute respiratory distress syndrome (ARDS), are life-threatening diseases characterized by diffuse pulmonary alveolar and interstitial damage that result in non-cardiogenic pulmonary edema, severe hypoxemia, and respiratory failure. Despite concerted efforts and extensive research that have sought to identify means of reducing the incidence of ALI, it is still associated with high rates of morbidity and mortality [1,2]. The pathogenesis of ALI is regarded as an acute inflammatory process caused by a range of local and systemic stimuli. The distinctive features of ALI include increased alveolar capillary permeability, accumulation and activation of neutrophil infiltration, upregulation of proinflammatory cytokines and transcription factors, and the generation of reactive oxygen species (ROS), which contribute to a further enhancement of alveolar–capillary barrier dysfunction and pulmonary cell death [3,4,5].

ALI is often induced by aspiration, shock, blood transfusion, or sepsis, and although the multiple risk factors and causes have yet to be sufficiently characterized, bacterial sepsis has been established to be the main cause of ALI in humans. Clinical research has indicated that inhalation of lipopolysaccharide (LPS), derived from the outer membrane of Gram-negative bacteria, is a potential causal factor, and this has commonly been used to elicit sepsis-induced ALI in animal models [6]. LPS is a strong trigger of lung inflammation that induces the innate immune system, accumulation of inflammatory cells, release of proinflammatory cytokines, and multiple signal pathways [7,8], and the findings of previous studies have indicated that the extracellular signal-regulated kinase (ERK) signaling pathway, one of the MAPK superfamily pathways, participates in the regulation of inflammation and oxidative stress [9]. Accordingly, the inhibition of ERK using specific inhibitors has been demonstrated to offer effective protection against LPS-induced ALI [10]. Moreover, the finding of recent studies have also indicated that nuclear factor kappa B (NF-*κ*B), a key nuclear transcription factor, plays an important role in oxidative stress and inflammatory processes. LPS can promote the activation of NF-*κ*B and the up-regulated expression of inflammatory cytokines and mediators [11,12]. Consistently, the inhibition of ERK and NF-*κ*B signaling pathways has been found to attenuate lung inflammation in LPS-induced ALI [13].

As further evidence indicating that the generation of ROS contributes to the initiation of endothelial damage and aggravates inflammation, it has been found that ROS-producing enzymes, nicotinamide adenine dinucleotide phosphate (NADPH) oxidases (NOXs), which catalyze the reduction of molecular oxygen (O_2_) to superoxide (O_2_^−^) play vital roles in the pathogenesis of ALI and ARDS [14,15]. Among these, the isoform NOX2, which is primarily expressed in phagocytes such as macrophages and neutrophils in lungs [16], promotes the uncontrolled generation of ROS implicated in the pathogenesis of several inflammatory disorders, and recent studies have demonstrated that NOX2-ROS signaling in neutrophils promotes TNF-α-induced inflammation in ALI [17]. Moreover, it has been established that neutrophil NOX2-induced acute inflammatory responses are mediated via NF-*κ*B activation [18].

Corilagin is a major polyphenolic compound obtained from the annual perennial plant *Phyllanthus urinaria*, which is used in traditional herbal medicine and has been established to have anti-inflammatory, anti-oxidative, and anti-apoptotic properties [19,20]. In a cellular model, corilagin has been demonstrated to down-regulate TNF-α expression via inhibition of the NF-*κ*B pathway [21], and also shown to have protective effects against LPS-induced liver injury via the suppression of oxidative stress [22]. In addition, this extract is found to reduce damage at the junctions of lung epithelial cells attributable to cigarette smoke [23] and attenuates aerosol bleomycin-induced lung injury via its anti-oxidative and anti-inflammatory activities [24]. These findings accordingly provide evidence to indicate that corilagin may play vital roles in inflammatory reactions and oxidative stress. To date, however, the pharmacological effects of corilagin in ALI have yet to be investigated. In this study, using a mouse model, we accordingly sought to evaluate the effects and underlying mechanisms of corilagin in LPS-induced ALI.

## 2. Materials and Methods

### 2.1. Animals

The C57BL/6 (B6) mice used in this study were purchased from BioLASCO Taiwan Co., Ltd. (Taipei, Taiwan). All animal procedures used in this study were approved by the Institutional Animal Care and Use Committee of Chang Gung Memorial Hospital, and all animal experiments were performed in accordance with the guidelines of the *Animal Welfare Act and the Guide for Care and Use of Laboratory Animals* of the National Institutes of Health. Animals were housed under environmentally controlled conditions (12 h light and dark cycle) and fasted overnight prior to commencing experimentation.

### 2.2. Cell Culture

Cells of the human lung type II epithelial cell line A549, obtained from the American Tissue Culture Collection (ATCC, Rockville, MD, USA), were cultured in DMEM culture medium (Sigma-Aldrich, Co., St. Louis, MO, USA) supplemented with 10% (*v/v*) fetal bovine serum (Invitrogen Co., Carlsbad, CA, USA) and 1% antibiotic–antimycotic solution (Gibco, #15240062). The cells were maintained in a CO_2_-enriched incubator at 37 °C in a humidified atmosphere containing 5% CO_2_, with the culture medium being replaced at 2-day intervals.

### 2.3. Cell Viability Assay

Cell viability was determined using Cell Counting Kit-8 (CCK-8) assay. Briefly, cells were seeded in 96-well plates at an initial density of 1 × 10^4^ cells/well, left overnight to adhere, and then treated with different doses of corilagin (Sigma Chemical Co., St. Louis, MO, USA) (0.1, 0.5, 1, 5, 10, or 20 µM) for 24 h. Following incubation, 10 μL of the CCK-8 solution was added to each well followed by incubation for 2–4 h. The optical density of well content was measured at 450 nm and cell viability was expressed as a percentage relative to that of the untreated control cells.

### 2.4. Experimental Procedures and Drug Treatment

Mice were randomly assigned to one of the following six groups (*n* = 6 per group): control (saline), corilagin (10 mg/kg), LPS (Sigma Chemical Co., St. Louis, MO, USA), LPS + dexamethasone (5 mg/kg), LPS + corilagin (5 mg/kg), and LPS + corilagin (10 mg/kg). Mice in the control and corilagin-only groups received an initial intratracheal instillation of 50 μL of phosphate-buffered saline (PBS) for 30 min, followed by intraperitoneal injection of saline or corilagin (10 mg/kg body weight), respectively. To induce ALI, mice in each of the treatment groups received intratracheal instillation of 2.5 μg/g LPS in 50 μL PBS, and following 30 min of LPS challenge, the mice were administered dexamethasone (5 mg/kg body weight), corilagin (5 or 10 mg/kg body weight), and an equal volume of saline, according to the aforementioned group designations. At 6 h after LPS administration, the animals were anesthetized and sacrificed, and lung tissues were immediately harvested for subsequent analyses.

### 2.5. Histology Analysis of Lung Tissues

To determine histopathological changes, harvested lung tissues were fixed with 4% paraformaldehyde, embedded in paraffin and sliced. Sections of 4 μm thickness were stained with hematoxylin and eosin (H&E) and subsequently observed under a DM2500 light microscope (Leica, Wetzlar, Germany).

### 2.6. Immunohistochemistry on Lung Tissues

Lung sections were dewaxed and blocked with blocking buffer for 30 min, after which they were incubated in primary antibodies against Ly6G and NOX2 (BD Biosciences Pharmingen, San Diego, CA, USA). Following incubation, the sections were washed with PBS for 5 min, and were thereafter labeled with biotin and streptavidin horseradish peroxidase-conjugated secondary antibodies according to the manufacturer’s instructions (Millipore IHC select kit). The reaction times for all samples were identical and having been processed these were inspected under a light microscope.

### 2.7. Measurement of Cytokine Levels in Lung Tissues

Lung tissue homogenates were used to determine the expression of TNF-α, IL-1β, and IL-6. The tissues were initially homogenized on ice, centrifuged at 12,000× *g* for 10 min at 4 °C, and the resulting supernatants were assayed using eBiosciences ELISA Kits (San Diego, CA, USA). Assays for each of the target cytokine were conducted in accordance with the instructions provided by the manufacturer, with absorbance in each case being determined at 450 nm. The levels of TNF-α, IL-1β, and IL-6 were standardized with respect to tissue weight.

### 2.8. Measurement of Tissue Glutathione and Malondialdehyde Levels

Lung tissues were homogenized in ice-cold 10% trichloroacetic acid and then centrifuged at 1000× *g* for 15 min at 4 °C. The supernatants thus obtained were subsequently re-centrifuged at 35,000× *g* at 4 °C for 8 min. Levels of glutathione (GSH) activity were determined using a glutathione assay kit (Cayman Chemical Company, MI, USA) and those of generated malondialdehyde (MDA), used as the indicator of lipid peroxidation, were determined using a Bioxytech MDA-586 Kit (OxisResearch, Portland, OR, USA). All procedures were performed according to the manufacturer’s protocols.

### 2.9. Western Blot Analysis

Lung tissues were initially lysed in buffer (1% Triton X-100, 10% glycerol, 20 mM Tris–HCl, 137 mM NaCl, 2 mM EDTA, 2 mM sodium pyrophosphate, 1 mM phenylmethylsulfonylfluoride, 0.1 ug/mL leupeptin, and 1 ug/mL aprotinin) and the cell lysates were centrifuged at 12,000× *g* for 10 min. Having collected the resulting supernatants, 40 μg aliquots of protein from each group were size-fractionated by sodium dodecyl sulfate polyacrylamide gel electrophoresis (SDS-PAGE) and transferred to polyvinylidene fluoride membranes (Schleicher & Schuell, Middlesex, UK) using a Hoefer electrophoresis and electrotransfer system (Amersham Biosciences, UK). After blocking with 5% fat-free milk solution and rinsing with Tris-buffer containing 1% Tween 20, the membranes were incubated with primary antibodies (1:1000) against ERK, phospho-ERK, NF-*κ*B, phospho- NF-*κ*B, and NOX2 (Cell Signaling Technology, MA, USA). After washing with TBS-T, the membranes were incubated with horseradish peroxidase-conjugated secondary antibodies. Labeled proteins were detected using an enhanced chemiluminescence system (Amersham, Piscataway, NJ, USA).

### 2.10. Statistical Analysis

Statistical analyses were performed using GraphPad Prism 6.0 Software (GraphPad Software Inc., San Diego, CA, USA). Comparisons between experimental groups were conducted using an one-way analysis of variance (ANOVA) with Tukey–Kramer multiple comparison tests. The experimental results are expressed as mean ± standard error of the mean (SEM). Statistical significance was defined as a *p* value *<* 0.05.

## 3. Results

### 3.1. Effects of Corilagin on A549 Cell Viability

We initially examined the effect of corilagin on the viability of A549 cells, which had been treated with increasing doses of corilagin (0–20 μM) for 24 h. As indicated in Figure 1, exposure to corilagin, appeared to have no significant cytotoxic effects on cell viability.

### 3.2. Effects of Corilagin on Histopathological Changes in LPS-Induced ALI

To characterize the pathological changes associated with LPS-induced ALI, we examined the histology of H&E-stained lung tissues. As shown in Figure 2, the lung tissues obtained from the control and corilagin-only group mice were characterized by a normal structure with no obvious pathological change (Figure 2A,B). In contrast, at 6 h after LPS induction, the tissues showed extensive histopathological changes, thickening of the alveolar walls, interstitial edema, and stromal hemorrhage (Figure 2C). After treatment with corilagin (5 and 10 mg/kg), these histopathological responses were significantly attenuated (Figure 2E,F). Similarly, treatment with dexamethasone (5 mg/kg) in the positive control group also significantly ameliorated LPS-induced lung damage (Figure 2D).

### 3.3. Effects of Corilagin on Neutrophil Infiltrations in LPS-Induced ALI

To observe the infiltration and accumulation of neutrophils in ALI, we performed immunohistochemical staining of lung tissue using antibodies against Ly6G, a granulocyte-specific marker. We accordingly found that compared with control group mice, tissues collected from the LPS-only-treated mice showed significant neutrophil infiltration in the vicinity of lesion site (Figure 3C). In contrast, lung tissues obtained from mice treated with dexamethasone (5 mg/kg) or corilagin (5 and 10 mg/kg) following LPS-induced lung injury were characterize by significantly lower neutrophil accumulation (Figure 3D–F).

### 3.4. Effects of Corilagin on Inflammatory Cytokine Production in Lung Tissues

To assess the effects of corilagin on inflammation, we examined the expression of the essential inflammatory cytokines TNF-α, IL-1β, and IL-6 in mice with LPS-induced ALI. We observed that compared with the control group, mice exposed to LPS for 6 h were characterized by significantly enhanced levels of these three cytokines (*p* < 0.005) (Figure 4). However, treatment with dexamethasone (5 mg/kg) was found to significantly attenuate the production of TNF-α and IL-1β (*p* < 0.005) compared with that observed in LPS-only-treated mice. Moreover, the lung tissues of mice administered 5 mg/kg corilagin were found to have significantly lower TNF-α, IL-1β, and IL-6 levels (*p* < 0.005, *p* < 0.01, and *p* < 0.005, respectively), and those mice receiving a higher dose (10 mg/kg) showed marked inhibition of these three inflammatory mediators (*p* < 0.005).

### 3.5. Effects of Corilagin on GSH and MDA Production in Lung Tissues

ALI is characterized by the generation of ROS and modulation of the antioxidant defense mechanism, with excess ROS production reducing the activity of antioxidant defense enzymes. In this context, we observed that the administration of LPS had the effect of significantly reducing the levels of GSH compared with those in the control group (*p* < 0.005) (Figure 5A), whereas treatment with corilagin (5 and 10 mg/kg) significantly reversed and increased GSH levels compared with those detected in the lungs of LPS group mice (*p* < 0.005). With respect MDA, used as a marker of lipid peroxidation, we found that compared with the control group, mice exposed to LPS were characterized by a significant increase in pulmonary MDA concentrations (Figure 5B). Conversely, following treatment with corilagin, we detected a significant reduction in MDA levels compared with those observed in the LPS-only-treated mice (*p* < 0.05). These findings thus tend to indicate that corilagin may have protective effects against LPS-induced oxidative stress injury.

### 3.6. Effects of Corilagin on Lung NOX2 Expression in LPS-Induced ALI

To further examine the inflammatory mechanisms associated with ALI, lung tissues were immunohistochemically stained with NOX2 antibodies. As shown in Figure 6A, compared with the control group, NOX2 expression was significantly higher in the lung tissues of mice in the LPS group. Contrastingly, corilagin administered at a dose of 5 mg/kg following LPS induction was found to promote a significant reduction in NOX2 expression in the vicinity of the inflamed lung tissues, and this inhibitory effect was further enhanced by treatment with dexamethasone (5 mg/kg) and a higher dose of corilagin (10 mg/kg). These observations were confirmed by Western blot analysis, which revealed that the NOX2 activity was significantly higher in the LPS group mice than that in mice of the control group (*p* < 0.01) (Figure 6B), and that mice treated with dexamethasone and 10 mg/kg corilagin were characterized by significantly lower levels of NOX2 expression compared with those in the LPS group (*p* < 0.05).

### 3.7. Effects of Corilagin on ERK and NF-*κ*B Expression in LPS-Induced ALI

In order to examine the anti-inflammatory mechanisms of corilagin, we investigated MAPK (ERK, JNK, and p38) and NF-*κ*B proteins in lung tissues using Western blot analysis, the results of which revealed a significant increase in the phosphorylated activated form of ERK at 6 h following LPS induction compared with that observed in the control group (*p* < 0.005) (Figure 7A). Corilagin treatment (5 and10 mg/kg) markedly inhibited the expression of p-ERK (*p* < 0.005), although it had no significant effect on the expression of JNK and p38 proteins in the lung (data not shown). With respect to a further intracellular signaling pathway protein, NF-*κ*B, we observed that compared with the control group, there was a significant increase in the expression of phosphorylated NF-*κ*B in the lungs in LPS-treated mice (*p* < 0.005; Figure 7B), whereas treatment with 10 mg/kg corilagin following LPS-induced lung injury effectively inhibited the phosphorylation of NF-*κ*B (*p* < 0.05).

## 4. Discussion

ALI and ARDS can cause life-threatening respiratory failure and are associated with high morbidity and mortality. Nevertheless, despite efforts to reduce the incidence of ALI, the numbers of respiratory failure cases reported in clinical settings remain a serious health problem. In the present study, we investigated the protective effect of corilagin, a major compound purified from *Phyllanthus urinaria*, using a mouse model of LPS-induced ALI. Our finding revealed that treatment with corilagin can significantly reduce LPS-induced acute inflammatory responses, including histopathological changes, neutrophil infiltration, the production of pro-inflammatory cytokines, and intracellular signal transduction parameters (i.e., ERK and NF-*κ*B), in lung tissues. In addition, we found that the administration of corilagin, 30 min after LPS challenge, suppressed oxidative-stress-related changes and lung NOX2 expression.

ALI is characterized by an acute pulmonary inflammatory reaction and infiltration of immune cells that contribute to diffuse alveolar and interstitial damage, pulmonary edema, parenchyma injury, and subsequent respiratory failure [1,25]. In the animal model used in the present study, we used intratracheally instilled LPS, an endotoxin derived from Gram-negative bacteria, to induce sepsis-related ALI. Within the lungs, the pulmonary innate immune system plays a significant role in the pathogenesis of ALI, in which pathogen-associated molecular pattern molecules, such as LPS, are recognized by resident alveolar macrophages, which mediate the early stages of the inflammatory response in ALI [26,27]. Upon activation, these cells release pro-inflammatory mediators and cytokines to recruit other monocytes and neutrophils into lesioned areas of the lung, thereby eliciting a subsequent inflammatory cascade and thus exacerbating the severity of ALI [4,28]. It has previously been established that LPS-induced ALI can be reduced by inhibiting the secretion of these cytokines and neutrophil recruitment [29,30], whereas administration of a selective antibody targeting the TNF-α receptor has been shown to similarly attenuate alveolar neutrophil recruitment and pulmonary inflammation in a non-human primate model of ALI [31]. Furthermore, the findings of in vitro human studies have indicated that corilagin has strong immunomodulatory effects on neutrophils [32]. In the present study, we found that the administration of LPS markedly enhanced the production of pulmonary cytokines, including TNF-α, IL-1β, and IL-6, neutrophil infiltration, and edema in lung tissues, and that subsequent treatment with corilagin significantly attenuated cytokine secretion, neutrophil infiltration, and lung parenchyma injury. These observations thus indicate corilagin has a potentially broad range of protective effects against LPS-induced ALI.

There is increasing evidence to indicate that oxidative stress, attributable to an imbalance between excess ROS production and activity of the antioxidant defense system, may contribute to ALI/ARDS. For example, it has been established that such stress can cause endothelial and epithelial barrier dysfunction in the lungs, thereby resulting in enhanced neutrophil transmigration and penetration across blood vessels into the inflamed tissues [33]. Further studies have revealed that corilagin might exert hepatoprotective effects by reducing acetaminophen-triggered ROS over-production and cytotoxicity [34], whereas in a model of bleomycin-induced lung injury, corilagin has been shown to dose-dependently reduce the development of oxidative stress [24]. In the present study, we established that exposure to LPS can promote a significant increase in the levels of MDA, which is taken to be indicative of an excess accumulation of ROS and associated lipid peroxidation. However, in response to corilagin treatment, we detected a notable reduction in the LPS-induced higher levels of MDA. Moreover, corilagin treatment was found to inhibit the LPS-induced depletion of GSH, an internal antioxidative enzyme that plays protective roles, reflecting the ability to scavenge harmful free radicals. Collectively, these findings indicate that corilagin may reduce LPS-induced ALI via its anti-oxidant effects.

There is growing evidence to indicate that the pathobiology of ALI/ARDS is associated with the enzymic generation of ROS, including the activation of NOX enzymes. Upon stimulation, NOX2, which is expressed primarily in phagocytes, such as alveolar macrophages and neutrophils, induces an enhancement of ROS production, thereby promoting inflammatory processes and exacerbating lung injury [16]. Some studies have shown that the NADPH-oxidase-deficient (p47^phox^-/-, a regulatory subunit essential for NOX2 activation) mice are characterized by reductions in free radical formation, vascular leakage, neutrophil infiltration, and LPS-induced lung damage [35], whereas in a model of LPS-induced lung injury, use of a peptide inhibitor of NOX2 activation has been shown to markedly reduce ROS production and mortality [36]. Furthermore, recent studies have provided evidence indicating that NOX2 activation in neutrophils plays an important role in promoting TNFα-induced acute inflammation in the lungs [17], and that NOX2-ROS signaling contributes to TNFα-induced NF-*κ*B activation in endothelial cells [18]. These findings thus indicate that NOX2 is an important determinant in the induction of inflammatory responses and ALI. Consistently, in the present study, we showed that corilagin can reduce the levels of NOX2 expression and oxidative stress in LPS-induced ALI, thereby indicating that inhibition of the NOX2 protein may contribute to the protective effects of corilagin against LPS-induced ALI.

With respect to LPS-induced ALI, oxidative stress has been established to activate intracellular signaling pathways, including the MAPK and NF-kB pathways, which are intrinsic sequences of reactions associated with the control of cytokine production and inflammatory responses. Among the MAPK family proteins, ERK is known to be involved in the regulation of inflammatory mediators and apoptotic events. In a previously examined model of acute liver injury, the authors demonstrated that suppression of the ERK signaling pathway has protective effects, assumed to be associated with a reduction in ROS production [37]. Similarly, inhibition of the ERK pathway, which contributes to TNF-α production, has been shown to attenuate LPS-induced pulmonary inflammatory responses [10]. Furthermore, members of the MAPK family play roles in inducing the activation of NF-*κ*B, a key transcription factor associated with inflammation, thereby resulting in the secretion of pro-inflammatory mediators and subsequent lung damage. In response to activation, NF-*κ*B is translocated from the cytoplasm to the nucleus, wherein it mediates the transcription of pro-inflammatory genes [38]. Inhibition of the phosphorylation and activation of ROS-mediated NF-*κ*B p65 reduces the expression of pro-inflammatory cytokines (TNF-α, IL-1β, and IL-6) and thus LPS-induced ALI [39]. With respect to pulmonary fibrosis, it has previously been reported that corilagin exerts antioxidant and anti-inflammatory effects by inhibiting the NF-*κ*B signaling pathway and cytokine production [21]. The findings of recent studies have also indicated that corilagin can target the MAPK and NF-*κ*B signaling pathways, thereby inducing their anti-inflammatory effects [40]. Consistent with these findings, the results of Western blot analyses performed in the present study indicate that LPS promotes the expression of p-ERK and p-NF-*κ*B, and that the administration of corilagin effectively alleviates this phosphorylation, thereby reducing the expression of ERK and NF-*κ*B. These observations provide convincing evidence that corilagin can protect against LPS-induced ALI by suppressing ERK/NF-*κ*B-mediated inflammatory pathways.

In terms of structure, corilagin (β-1-O-galloyl-3,6-(R)-hexahydroxydiphenoyl-d-glucose) is a polyphenolic compound. In a previous study [41], the various bioactive compounds were isolated from peel of *Punica granatum* to evaluate their anti-oxidant activity. The results showed that 2,3-(S)-hexahydroxydiphenoyl-glucose had significant anti-oxidant potential among these compounds. The presence of anti-oxidant properties of hexahydroxydiphenoyl-glucose structure has been demonstrated. Further studies regarding the structure–activity relationship for corilagin and its effects are required. In addition, the structure of corilagin includes a polyphenolic motif. Other polyphenolic compounds such as tannic acid and curcumin also have similar anti-inflammatory and anti-oxidant effects. Tannic acid is a natural high molecular weight polyphenolic compound and may contribute to the effect of anti-inflammation by decreasing MPO enzyme activity in a rat model [42]. Curcumin in dietary supplements is also considered to have significant anti-oxidant and anti-inflammatory therapeutic effects [43].

## 5. Conclusions

In this study, we demonstrated that by attenuating oxidative stress and inflammatory activities, corilagin has protective effects against LPS-induced ALI in mice, the underlying mechanisms of which appear to involve reductions in the activities of NOX2 and ERK/NF-*κ*B signaling pathways. On the basis of these findings, we believe that corilagin has potential utility as a therapeutic agent for the treatment of this life-threatening respiratory disease.

## Figures and Tables

**Figure 1 biology-11-01058-f001:**
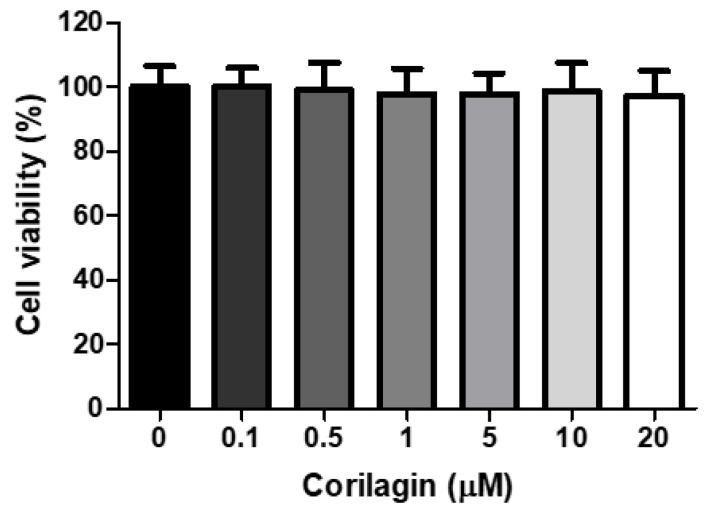
**Cell viability of A549 cells**. A549 cells were treated with different concentrations of corilagin (0, 0.1, 0.5, 1, 5, 10, and 20 μM) for 24 h. The results are presented as a percentage of the control and expressed as the mean ± SEM.

**Figure 2 biology-11-01058-f002:**
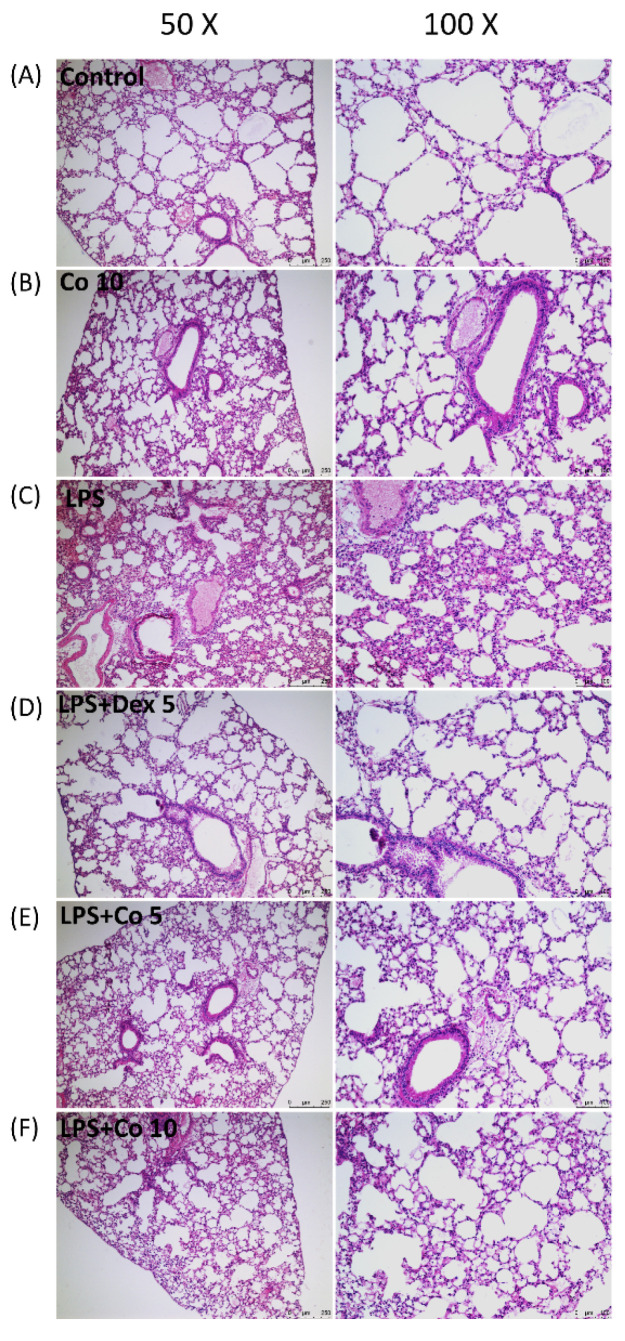
**Effects of corilagin on histopathological changes in lung tissues.** After intratracheal LPS challenge for 30 min, the mice were administered with Dex (5 mg/kg), corilagin (5 and 10 mg/kg), or equal volume of saline intraperitoneally. Mice were sacrificed after 6 h of LPS challenge. Representative histological changes of the lung tissues obtained from six groups: (**A**) normal saline (control), (**B**) corilagin (10 mg/kg) only, (**C**) LPS, (**D**) LPS + dexamethasone (5 mg/kg), (**E**) LPS + corilagin (5 mg/kg), and (**F**) LPS + corilagin (10 mg/kg). (Hematoxylin and eosin staining, 50× and 100× magnifications are shown).

**Figure 3 biology-11-01058-f003:**
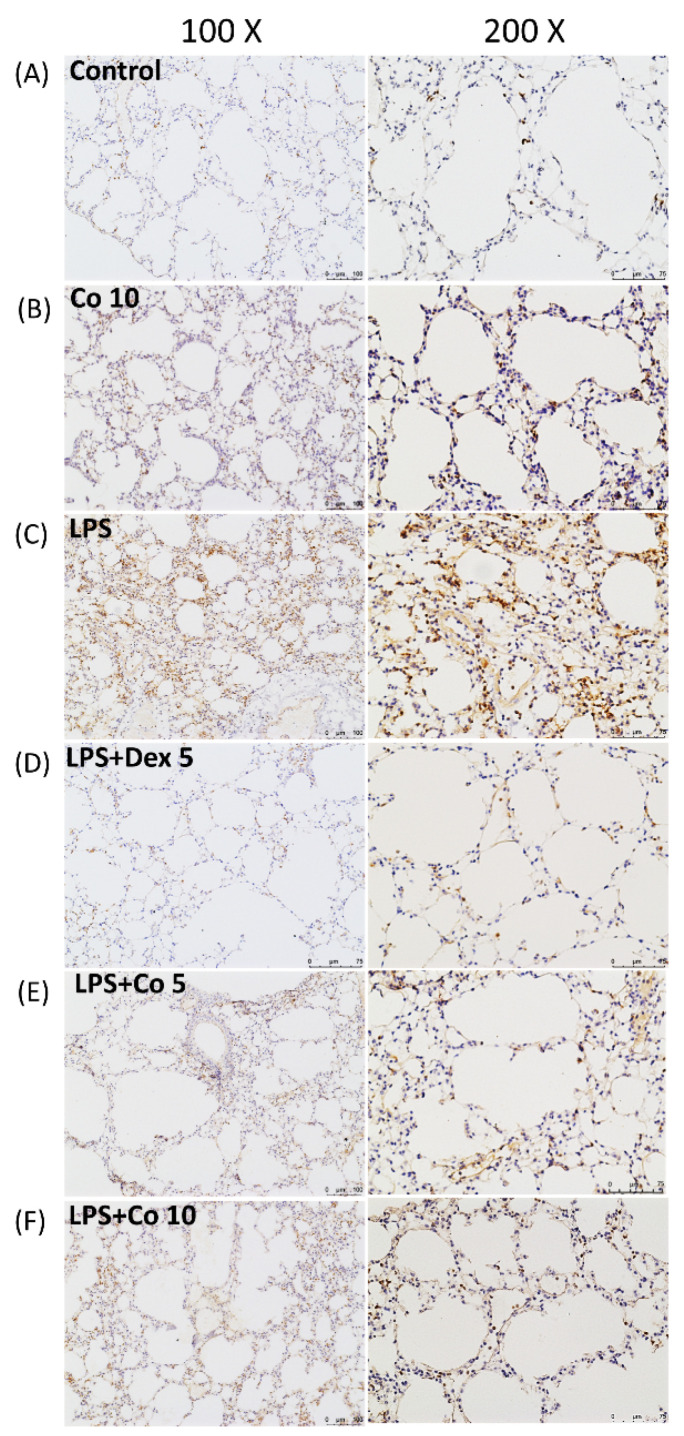
**Effects of corilagin on neutrophil infiltrations in LPS-induced ALI**. After intratracheal LPS challenge for 30 min, the mice were administered with Dex (5 mg/kg), corilagin (5 and 10 mg/kg), or equal volume of saline intraperitoneally. Mice were sacrificed after 6 h of LPS challenge for analysis by immunohistochemical staining. Lung tissues were immunostained with anti–Ly6G antibody (brown). Typical images were chosen from each group: (**A**) normal saline (control), (**B**) corilagin (10 mg/kg) only, (**C**) LPS, (**D**) LPS + dexamethasone (5 mg/kg), (**E**) LPS + corilagin (5 mg/kg), and (**F**) LPS + corilagin (10 mg/kg). (100× and 200× magnifications are shown).

**Figure 4 biology-11-01058-f004:**
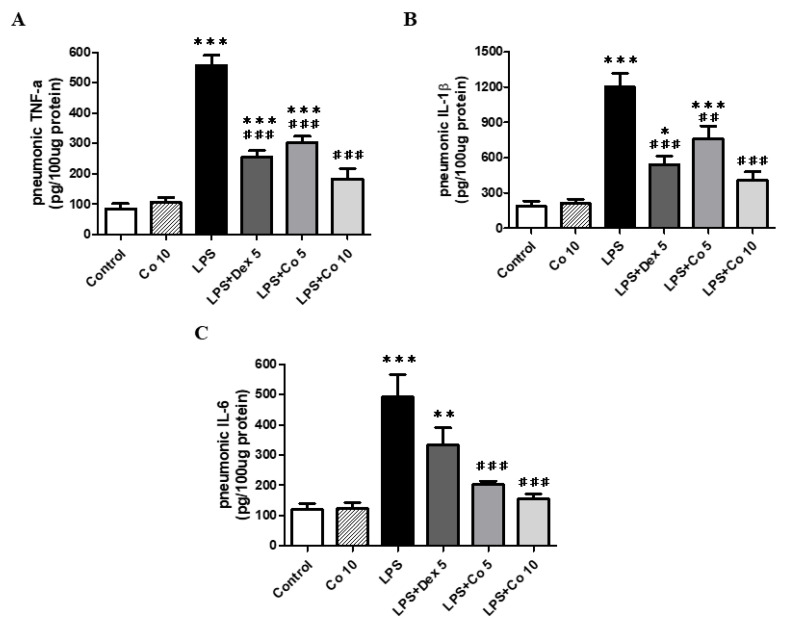
**Effects of corilagin on TNF-α (A), IL-1β (B), and IL-6 (C) expressions in lung tissues**. After intratracheal LPS challenge for 30 min, the mice were administered with Dex (5 mg/kg), corilagin (5 and 10 mg/kg), or equal volume of saline intraperitoneally. Mice were sacrificed after 6 h of LPS challenge for analysis of lung TNF-α, IL-1β, and IL-6 levels. Each value represents mean ± SEM of six mice per group. * *p* < 0.05, ** *p* < 0.01, *** *p* < 0.005 vs. control; ^##^ *p <* 0.01, ^###^ *p <* 0.005 vs. LPS alone.

**Figure 5 biology-11-01058-f005:**
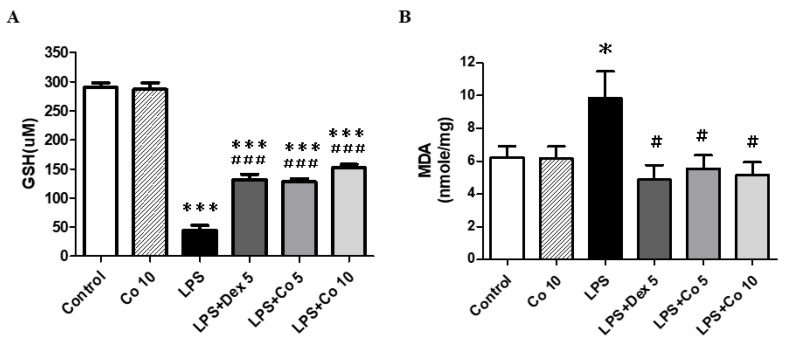
**Effects of corilagin on oxidative stress in LPS-induced ALI**. After intratracheal LPS challenge for 30 min, the mice were administered with Dex (5 mg/kg), corilagin (5 and 10 mg/kg), or equal volume of saline intraperitoneally. Mice were sacrificed, and lung tissues were obtained after 6 h of LPS challenge. The lung tissue (**A**) GSH and (**B**) MDA levels are expressed as means ± SEM (*n* = 6 mice/group). * *p* < 0.05, *** *p* < 0.005 vs. control; ^#^ *p* < 0.05, ^###^ *p* < 0.005 vs. LPS alone.

**Figure 6 biology-11-01058-f006:**
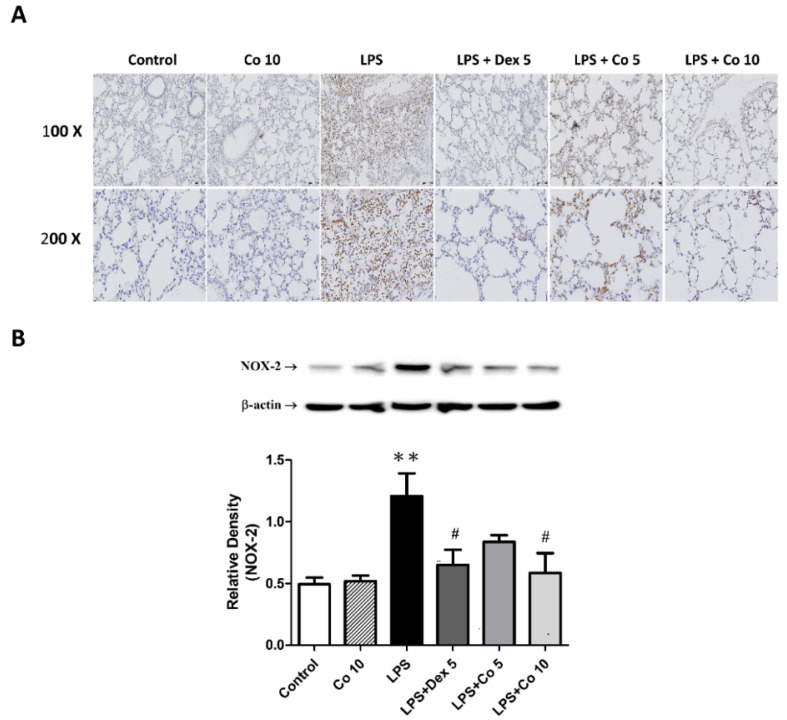
**Effects of corilagin on NOX2 expression in LPS-induced ALI**. After intratracheal LPS challenge for 30 min, the mice were administered with Dex (5 mg/kg), corilagin (5 and 10 mg/kg), or equal volume of saline intraperitoneally. Mice were sacrificed, and lung tissues were obtained after 6 h of LPS challenge. (**A**) Immunohistochemical staining of lung NOX2 expression (brown) from six groups. Typical images were chosen from each experimental group (100× and 200× magnifications are shown). (**B**) The expression levels of NOX2 were detected by Western blot analysis. The bands were analyzed using densitometry, and each value represents the mean ± SEM (*n* = 6 mice/group). ** *p* < 0.01 vs. control; ^#^ *p* < 0.05 vs. LPS alone.

**Figure 7 biology-11-01058-f007:**
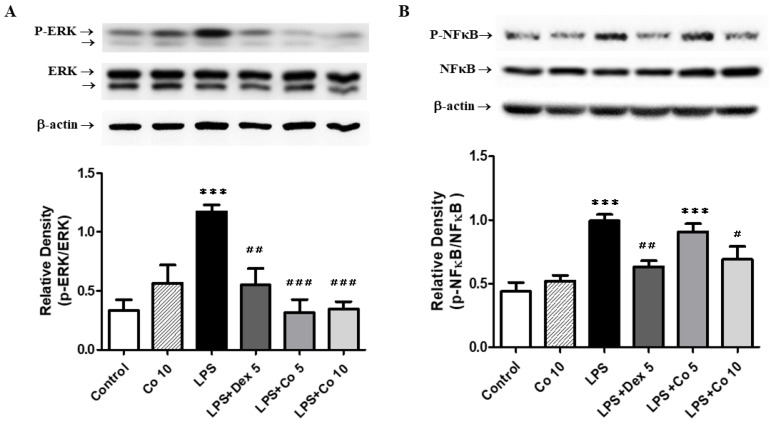
**Effects of corilagin on lung** (**A**) **ERK and** (**B**) **NF-*κ*B expression in LPS-induced ALI**. After intratracheal LPS challenge for 30 min, the mice were administered with Dex (5 mg/kg), corilagin (5 and 10 mg/kg), or equal volume of saline intraperitoneally. Mice were sacrificed, and lung tissues were obtained after 6 h of LPS challenge. The expression levels of ERK and NF-*κ*B were detected by Western blot analysis. Equal protein loading is illustrated by the β-actin bands in all lanes. The bands were analyzed using densitometry, and each value represents mean ± SEM of six mice per group. *** *p* < 0.005 vs. control; ^#^ *p* < 0.05, ^##^ *p* < 0.01, ^###^ *p <* 0.005 vs. LPS alone.

## Data Availability

The data presented in this study are available on request from the corresponding author.

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
