# Peer review of "Effects of Corilagin on Lipopolysaccharide-Induced Acute Lung Injury via Regulation of NADPH Oxidase 2 and ERK/NF-κB Signaling Pathways in a Mouse Model"

_biology, 2022, doi:10.3390/biology11071058_

Round 1

Reviewer 1 Report

In the manuscript of Liu et al., entitled “Effects of corilagin on lipopolysaccharide-induced acute lung 2 injury via regulation of NADPH oxidase 2 and ERK/NF-?B signaling pathways in a mouse model” the author evaluated the protective effects of corilagin and showed the mechanisms on LPS induced ALI. This is a very well-articulated article. However, some concerns needed to be addressed before publication. Please find below the minor concerns of this study.

Comments:

1.     What were the criteria for selecting 5mg or 10mg/kg body weight for the study? The author should write “5 mg or 10mg/kg body weight” instead of “5mg or 10mg/kg”

2.     In this study, Corilagin showed an anti-inflammatory effect.  However, the statement in Line 422, indicated the opposite information. Thus the author should correct these contradictory statements.

3.     For the neutrophil infiltration study, I would suggest the author to collect BALF of the mice and perform flow cytometry using neutrophil markers (Ly6G). This would provide more scientific and quantitative information that would give more impact on the article.

4.     The authors should add the compound name to Figures 2 and Figure 3.

Author Response

Revise [biology-1807042] entitled, “Effects of corilagin on lipopolysaccharide-induced acute lung injury via regulation of NADPH oxidase 2 and ERK/NF-kB signaling pathways in a mouse model

Dear Editor:

Thank you very much for your e-mail dated July 6, 2022. We appreciate your kind comments that the manuscript has merit and also your kind invitation to revise our manuscript. Thank you also for sending me the comments of the reviewers. The comments/suggestions of the reviewers were very valuable and we have made changes in the manuscript taking into account those comments/suggestions. I am therefore taking the liberty of listing the comments of the referees, our response to them as well as listing the changes that have been made in the accompanying pages. The changes in the revised manuscript are highlighted.

We sincerely hope that changes made in the revised manuscript meet with your approval and the approval of the referees and that our manuscript is now acceptable for publication in your esteemed journal. If there are further questions, please do not hesitate to contact me.

Sincerely yours,

Huang-Ping Yu, MD, PhD

Department of Anesthesiology, Chang Gung Memorial Hospital and Chang Gung University College of Medicine, No. 5, Fushing 1st Rd, Gueishan, Taoyuan 33305, Taiwan.

Telephone: +886-3-3281200 ext-3624;

Fax: +886-3-3281200 ext-2787

Response to Reviewer 1

Reviewer 1's comments:

In the manuscript of Liu et al., entitled “Effects of corilagin on lipopolysaccharide-induced acute lung 2 injury via regulation of NADPH oxidase 2 and ERK/NF-?B signaling pathways in a mouse model” the author evaluated the protective effects of corilagin and showed the mechanisms on LPS induced ALI. This is a very well-articulated article. However, some concerns needed to be addressed before publication. Please find below the minor concerns of this study.

R:

We greatly appreciate the kind comments that our manuscript is a very well-articulated article. Our responses to reviewer’s kind comments are as follows.

Comments

  1. What were the criteria for selecting 5mg or 10mg/kg body weight for the study? The author should write “5 mg or 10mg/kg body weight” instead of “5mg or 10mg/kg”

R1:

We thank the reviewer for this kind comment. We selected 5mg or 10mg/kg body weight for the study due to significant therapeutic effects to reduce acute lung injury. However, there is no obvious effect when the mice were administered with less than 5mg/kg body weight of corilagin according to our preliminary data.       

Besides, we have changed “5mg or 10mg/kg” to “5 mg or 10mg/kg body weight” in our revised manuscript.

In the Abstract section that read “ALI was induced in mice by the intratracheal administration of LPS, and following 30 min of LPS challenge, corilagin (5 and 10 mg/kg body weight) was administered intraperitoneally.……” (see Abstract: line 22);

In the Materials and Methods section that read “2.4 Experimental procedures and drug treatment: …… followed by intraperitoneal injection of saline or corilagin (10 mg/kg body weight), respectively. To induce ALI, mice in each of the treatment groups received intratracheal instillation of 2.5 μg/g LPS in 50 μL PBS, and following 30 min of LPS challenge, the mice were administered dexamathasone (5 mg/kg body weight), corilagin (5 or 10 mg/kg body weight)……” (see Materials and Methods: line 116 to line 119).

  1. In this study, corilagin showed an anti-inflammatory effect. However, the statement in Line 422, indicated the opposite information. Thus the author should correct these contradictory statements.

R2:

We greatly appreciate the reviewer for this suggestion. We correct these words in the manuscript.

In the Conclusion section that read “In this study, we demonstrated that by attenuating oxidative stress and inflammatory activities, corilagin has protective effects……” (see Conclusion: line 401 to line 402).

  1. For the neutrophil infiltration study, I would suggest the author to collect BALF of the mice and perform flow cytometry using neutrophil markers (Ly6G). This would provide more scientific and quantitative information that would give more impact on the article.

R3:

We thank the reviewer for this suggestion. In this study, immunohistochemical staining of lung tissue showed significant neutrophil infiltration in the lesion site after LPS challenge. We will consider to collect cellular samples from BALF and perform flow cytometry in the future analysis.

  1. The authors should add the compound name to Figure 2 and Figure 3.

R4:

We greatly appreciate the reviewer for this comment. We added the compound name in Figures 2 and 3. The results are showed in the revised Figure 2 and Figure 3. 

Reviewer 2 Report

This submitted manuscript (biology-1807042) was reveal that corilagin was able to protect inflammatory reaction and infiltration of immune cell induced by LPS on in vivo mouse model. And then, the activity was suggested via ERK/NF-kB signaling pathway. These knowledges were attractive information for development of novel therapeutic agent for ALI and ARDS. Since the methods, results and strategy for this research project was suitable and clearly, I had a good feeling for this manuscript.

The structure of corilagin was included polyphenolic motif. There are so many polyphenolic compounds in the world such as tannic acid, curcumin and so on. Did other polyphenolic compound have similar effects? Would you investigate the structure-activity relationship for corilagin?

The used reagent, corilagin, was purchased from sigma-chemical. So, it’s chemical pure compound but not crude extract. The description of “extract” should be altered to compound or reagent.

At finally, the supplied company name and grade of LPS should be reveal in text.

I hope that these comments will be helpful for your research.

Author Response

Response to Reviewer 2

Reviewer 2's comments:

This submitted manuscript (biology-1807042) was reveal that corilagin was able to protect inflammatory reaction and infiltration of immune cell induced by LPS on in vivo mouse model. And then, the activity was suggested via ERK/NF-kB signaling pathway. These knowledges were attractive information for development of novel therapeutic agent for ALI and ARDS. Since the methods, results and strategy for this research project was suitable and clearly, I had a good feeling for this manuscript.

R:

We greatly appreciate the reviewer for these kind comments that these knowledges are attractive information and the methods, results and strategy for this research project is suitable and clearly.

  1. The structure of corilagin was included polyphenolic motif. There are so many polyphenolic compounds in the world such as tannic acid, curcumin and so on. Did other polyphenolic compound have similar effects? Would you investigate the structure-activity relationship for corilagin?

R1:

We thank the reviewer for this kind comment. Corilagin (β-1-O- galloyl-3,6-(R)- hexahydroxydiphenoyl-d-glucose) is a polyphenolic compound. It has been documented to possess

many pharmacological properties, such as antioxidant, anti-inflammatory, hepatoprotective, and anti-tumor

activities. Other polyphenolic compounds such as tannic acid, curcumin have similar anti-inflammatory effects. Tannins are high molecular weight phenolic compounds and may contribute to the treatment of inflammation

by decreasing MPO enzyme activity in a rat model (Ref. 1). Curcumin curcuminoids are also considered to have

anti-oxidant and anti-inflammatory curcumin dietary supplements (Ref. 2).

    Corilagin is an ellagitannin with a hexahydroxydiphenoyl group. In a previous study (Ref. 3), the different bioactive compounds were isolated from peel of Punica granatum and to evaluate the antioxidant activity of various extracts. The results showed that 2,3‑(S)‑hexahydroxydiphenoyl‑glucose had significant antioxidant and radical quenching potential among these compounds. The presence of antioxidant properties of 2,3‑(S)‑hexahydroxydiphenoyl‑glucose has been demonstrated. Further studies regarding the structure-activity relationship for corilagin and its effects are required to be explored.

References

Ref. 1: Soyocak, A.; Kurt, H.; Cosan, D.T.; Saydam, F.; Calis, I.U.; Kolac, U.K.; Koroglu, Z.O.; Degirmenci, I.; Mutlu, F.S.; Gunes, H.V. Tannic acid exhibits anti-inflammatory effects on formalin-induced paw edema model of inflammation in rats. Hum Exp Toxicol 2019, 38, 1296-1301, doi:10.1177/0960327119864154.

Ref. 2: Peng, Y.; Ao, M.; Dong, B.; Jiang, Y.; Yu, L.; Chen, Z.; Hu, C.; Xu, R. Anti-Inflammatory Effects of Curcumin in the Inflammatory Diseases: Status, Limitations and Countermeasures. Drug Des Devel Ther 2021, 15, 4503-4525, doi:10.2147/DDDT.S327378.

Ref. 3: Jacob, J.; Lakshmanapermalsamy, P.; Illuri, R.; Bhosle, D.; Sangli, G.K.; Mundkinajeddu, D. In vitro Evaluation of Antioxidant Potential of Isolated Compounds and Various Extracts of Peel of Punica granatum L. Pharmacognosy Res 2018, 10, 44-48, doi:10.4103/pr.pr_36_17.

In view of this, we have added additional information in the Discussion section and Reference in our revised manuscript.

In the Discussion section that read “In terms of structure, corilagin (β-1-O-galloyl-3,6-(R)-hexahydroxydiphenoyl-d-glucose) is a polyphenolic compound. In a previous study [41], the various bioactive compounds were isolated from peel of Punica granatum and to evaluate the anti-oxidant activity. The results showed that 2,3‑(S)‑hexahydroxydiphenoyl‑glucose had significant anti-oxidant potential among these compounds. The presence of anti-oxidant properties of hexahydroxydiphenoyl‑glucose structure has been demonstrated. Further studies regarding the structure-activity relationship for corilagin and its effects are required to be explored. In addition, the structure of corilagin was included polyphenolic motif. Other polyphenolic compounds such as tannic acid and curcumin also have similar anti-inflammatory and anti-oxidant effects. Tannic acid is a natural high molecular weight polyphenolic compound and may contribute to the effect of anti-inflammation by decreasing MPO enzyme activity in a rat model [42]. Curcumin is also considered to have significant anti-oxidant and anti-inflammatory therapeutic effects on dietary supplements [43].” (see Discussions: line 387 to line 399).

References

  1. Jacob, J.; Lakshmanapermalsamy, P.; Illuri, R.; Bhosle, D.; Sangli, G.K.; Mundkinajeddu, D. In vitro Evaluation of Antioxidant Potential of Isolated Compounds and Various Extracts of Peel of Punica granatum L. Pharmacognosy Res 2018, 10, 44-48, doi:10.4103/pr.pr_36_17.
  2. Soyocak, A.; Kurt, H.; Cosan, D.T.; Saydam, F.; Calis, I.U.; Kolac, U.K.; Koroglu, Z.O.; Degirmenci, I.; Mutlu, F.S.; Gunes, H.V. Tannic acid exhibits anti-inflammatory effects on formalin-induced paw edema model of inflammation in rats. Hum Exp Toxicol 2019, 38, 1296-1301, doi:10.1177/0960327119864154.
  3. Peng, Y.; Ao, M.; Dong, B.; Jiang, Y.; Yu, L.; Chen, Z.; Hu, C.; Xu, R. Anti-Inflammatory Effects of Curcumin in the In-flammatory Diseases: Status, Limitations and Countermeasures. Drug Des Devel Ther 2021, 15, 4503-4525, doi:10.2147/DDDT.S327378.

  1. The used reagent, corilagin, was purchased from sigma-chemical. So, it’s chemical pure compound but not crude extract. The description of “extract” should be altered to compound or reagent.

R2:

We greatly thank the reviewer for this suggestion. We changed “extract” to “compound” in our revised manuscript.

In the Abstract section that read “Corilagin, a major polyphenolic compound obtained from the herb Phyllanthus urinaria, has an-ti-inflammatory and antioxidant properties.……” (see Abstract: line 18);

In the Introduction section that read “Corilagin is a major polyphenolic compound obtained from the annual perennial plant Phyllanthus urinaria……” (see Introduction: line 73);

In the Discussion section that read “In the present study, we investigated the protective effect of corilagin, a major compound purified from Phyllanthus urinaria,……” (see Discussion: line 297).

  1. At finally, the supplied company name and grade of LPS should be reveal in text.

R3:

We thank the reviewer for this suggestion. We added company name and grade of LPS in our revised manuscript.

In the Materials and Methods section that read “Mice were randomly assigned to one of the following six groups (n = 6 per group): control (saline), corilagin (10 mg/kg), LPS (Sigma Chemical Co., St. Louis, MO, USA), LPS + dexamethasone……” (see Materials and Methods: line 112).

  1. I hope that these comments will be helpful for your research.

R4:

We greatly appreciate these kind comments from the reviewer. These kind comments are helpful for our research. Many thanks.
